# Improved Antioxidant Capacity of *Akebia trifoliata* Fruit Inoculated Fermentation by *Plantilactobacillus plantarum*, Mechanism of Anti-Oxidative Stress through Network Pharmacology, Molecular Docking and Experiment Validation by HepG2 Cells

**Yuhao Sun** [1]**, Zhenzhen Wang** [1]**, Jing Dai** [1] **, Ruyi Sha** [1,*]**, Jianwei Mao** [2]**, Yangchen Mao** [3] **and Yanli Cui** [4]

[1] School of Biological and Chemical Engineering, Zhejiang University of Science and Technology, Hangzhou 310023, China; independentsun@163.com (Y.S.); cnhkwzz@163.com (Z.W.); nanodoudou123@163.com (J.D.)

[2] School of Jianhu, Zhejiang Industry Polytechnic College, Shaoxing 312000, China; zjhzmjw@163.com

[3] School of Medicine, University of Southampton, Southampton SO17 1BJ, UK; m13735877964@163.com

[4] Department of Chemistry, Zhejiang University, Hangzhou 310027, China; cuiyl@zju.edu.cn

\* Correspondence: kevinsha_0204@163.com

**Abstract:** In this work, spontaneously fermented and inoculation-fermented *Akebia trifoliata* fruit Jiaosu (SFAJ/IFAJ) were compared. The key metabolites and antioxidant activities of SFAJ and IFAJ were tracked and tested during fermentation. The antioxidant effect of fermented *Akebia trifoliata* fruit and the underlying mechanisms were explored using network pharmacology for the prediction and verification of the molecular targets and pathways of the *Akebia trifoliata* fruit's action against oxidative stress. Furthermore, the results were verified by molecular docking and then investigated, based on a HepG2 cell model. The results of correlation analysis and principal component analysis (PCA) showed that there were significant positive correlations between the phenols, flavonoids, and terpenoids in SFAJ and IFAJ and their antioxidant activities. Network pharmacology and molecular docking analysis disclosed the antioxidation mechanism at the molecular level. In addition, both SFAJ and IFAJ were effective at alleviating oxidative stress in HepG2 cells. In particular, IFAJ performed better than SFAJ in protecting cells with an intracellular reactive oxygen species (ROS) level of $99.96 \pm 4.07$ U/mg prot, superoxide dismutase (SOD) activity of $41.56 \pm 0.06$ U/mg prot, catalase (CAT) activity of $91.78 \pm 3.85$ U/mg prot, and glutathione peroxidase (GSH-Px) activity of $39.32 \pm 2.75$ mU/mg prot in the IFAJ group. Collectively, this study revealed the changes in bioactive metabolite contents and the in vitro antioxidant activity during fermentation and investigated the protectiveness of SFAJ and IFAJ against oxidative stress within HepG2 cells, promoting the study of the antioxidant efficacy of IFAJ, thereby providing valuable reference data for the optimization of its preparation and the development of relevant products with health benefits.

**Keywords:** *Akebia trifoliata* fruit; bioactive metabolites; antioxidant activity; network pharmacology; molecular docking; oxidative stress

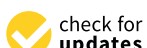



## 1. Introduction

*Akebia trifoliata*, which belongs to the family of Lardizabalaceae, genus *Akebia*, is a deciduous woody liana that is naturally distributed among mountainous and hilly areas in Asia, especially in China, Japan, and Korea [1]. *Akebia trifoliata* fruit have a sweet and juicy pulp containing multiple saccharides, organic acids, and vitamins [2], seeds rich in unsaturated fatty acids and proteins [3], suitable for oil and plant protein extraction, along with the pericarp and stem, which contain phenols, flavonoids, and terpenoids [1]. In addition, the *Akebia trifoliata* fruit is a traditional Chinese medicine possessing toxin

elimination, immunity boost, antitumor, anti-inflammatory, diuretic, and antibacterial effects, and other physiological or medicinal functions [4–8]. *Akebia trifoliata* fruits have numerous seeds wrapped by a thick layer of peel, yielding an extremely low edible rate of only 19–32%. The peel spontaneously cracks easily during the maturation period, causing great difficulties in the preservation of fruit. These drawbacks badly hinder its application in the food field. Limited by a short harvesting period and shelf life, the majority of the fresh fruit is further processed in food factories and sold as canned or bottled commodities in markets. As yet, *Akebia trifoliata* fruit has merely been used for the production of juice, wine, vinegar, jelly, dried fruit, and other highly processed products [9]. Currently, boiling, air-drying at high temperatures, extraction with organic solvents, and long-term soaking coupled with the excessive addition of artificial additives are the mainstream for fruit processing. During the processing, involving these complicated and outdated methods, a large portion of the fruit is discarded (pericarp, seeds, etc.) and the bioactive compounds are severely damaged, causing massive losses both in economic terms and to the physiological functions beneficial to human health.

Fermentation as an ancient technology has been used to enhance the shelf-life and nutritional qualities of vegetables, fruits, and legumes, increasing the content of bioactive phenolic compounds, thereby enhancing their antioxidant activity. The edible plant Jiaosu (EPJ) is recognized as a traditional fermented product in China and is fermented by microorganisms along with a variety of fruit, vegetables, edible fungi, and medicinal and edible traditional medicine as raw materials [10,11]. It is gaining increasing popularity in both East and Southeast Asian countries by virtue of its unique benefits to human health, as confirmed by a large body of literature [12–14]. Relevant enterprise standards and guidelines [15,16] have been released to encourage the popularization, industrialization, localization, and standardization of EPJ products. Spontaneous fermentation is often used to ferment EPJ; this is a method of traditional preservation and nutrition enhancement of fruit and vegetables, using the probiotic lactic acid bacteria (LAB) present in the plant tissues. Generally, the disadvantages of spontaneous fermentation are uncontrollable fermentation, ease of infection by bacteria, and inconsistent product quality. In comparison, inoculated fermentation can avoid the contamination of pathogenic bacteria, facilitate the regulation of fermentation, and obtain products of consistent quality. In addition, artificially inoculated fermentation can guarantee the same performance during the processing of EPJ. However, to the best of our knowledge, no reports have been published on the antioxidant effects and mechanisms of *Akebia trifoliata* fruit for the treatment of anti-oxidative stress until now.

In this study, *Akebia trifoliata* fruit was used as the base material to ferment into EPJ, and naturally fermented *Akebia trifoliata* fruit Jiaosu (SFAJ) and inoculation-fermented *Akebia trifoliata* fruit Jiaosu (IFAJ) inoculated with *Plantilactobacillus plantarum* were investigated simultaneously, in pursuit of improving the antioxidative activity of EPJ products. The changes in antioxidant metabolites and the correlation between bioactive metabolites and antioxidant activity were investigated in the context of SFAJ and IFAJ. Thus, the potential efficacy of the fermented *Akebia trifoliata* fruit Jiaosu to reduce the oxidative stress in HepG2 cells that is induced by $H_2O_2$ was revealed; the mechanism of action was explored by using network pharmacology, molecular docking analysis, and experiment validation. The study might provide a novel idea for the full use of *Akebia trifoliata* fruit and also lay the foundations for the development of relevant beverages with health-conducive functionalities or other high-value-added products.

## 2. Materials and Methods

### 2.1. Materials

*Akebia trifoliata* fruits were obtained from the Guilin seedling base (Guilin, China). Premium white granulated sugar was purchased from Taikoo Sugar Refinery Co., Ltd. (Shanghai, China). *Plantilactobacillus plantarum* JYLP-002 ($1 \times 10^{10}$ CFU/g) was purchased from Shandong Zhongke Jiayi Biological Engineering Co., Ltd. (Weifang, China). Pectinase (10,000 U/g) was purchased from Shandong Longkete Enzyme Preparation Co., Ltd. (Linyi, China).

### 2.2. Chemicals

Magnesium sulfate heptahydrate, phenol, and Folin–Ciocalteu reagent were purchased from Sinopharm Chemical Reagent Co., Ltd. (Shanghai, China). Sodium hydroxide, potassium hydroxide, sodium carbonate, sodium nitrite, aluminum nitrate, citric acid, sodium citrate, hydrochloric acid, sulfuric acid, 1,1-diphenyl-2-picrylhydrazyl (DPPH), tri-hytdroxymethylaminomethane (Tris), 2,2′-azino-bis-(3-ethylbenzthiazoline-6-sulphonate) (ABTS), potassium persulfate, sodium salicylate, and ferrous sulfate were purchased from Shanghai Ling Feng Chemical Reagent Co., Ltd. (Shanghai, China). Sodium dihydrogen phosphate, potassium dihydrogen phosphate, gallic acid, rutin, oleanolic acid, phosphoric acid, and methanol were purchased from Shanghai Aladdin Bio-chem Tech Co., Ltd. (Shanghai, China). HepG2 human hepatocellular carcinoma cells were kindly provided by the First Affiliated Hospital of the Medical College of Zhejiang University. Minimum essential medium (MEM), penicillin-streptomycin solution ($100\times$), phosphate buffer solution (PBS), fetal bovine serum (FBS), and trypsin-EDTA solution were purchased from Procell Life Science & Technology Co., Ltd. (Wuhan, China). The reactive oxygen species (ROS) assay kit, superoxide dismutase (SOD) assay kit, and RIPA lysis buffer (high intensity) were purchased from the Nanjing Jiancheng Bioengineering Institute (Nanjing, China). The CCK-8 cell viability assay kit, catalase assay kit (CAT), and total glutathione peroxidase (GSH-Px) assay kit with NADPH were purchased from Beyotime Biotechnology Co., Ltd. (Shanghai, China).

### 2.3. Preparation and Sample Collection of SFAJ and IFAJ

The preparation of EPJ with *Akebia trifoliata* fruit as the raw material was conducted, referring to previous studies but with modifications [14,17–19]. Briefly, fresh and ripe *Akebia trifoliata* fruits (including peels and seeds) were cleaned with sterile water, then air-dried and mashed into a pulp using a blender. A total of 2.5 kg of pulp, 2.5 kg of sterile water, and 7.5 g of pectinase were mixed well in a 10-L sterile stainless-steel fermenter with a tap, then enzymatic hydrolysis was carried out at 40 °C for 12 h. After that, 2.5 kg of white granulated sugar, 2.5 kg of sterile water, 3.75 g of *Plantilactobacillus plantarum*, 0.125 g of magnesium sulfate heptahydrate, and 0.125 g of sodium dihydrogen phosphate were added into the fermentation tank, mixed well, and fermented in an airtight manner at $25 \pm 5$ °C for 72 days. The product was nominated as inoculation-fermented *Akebia trifoliata* fruit Jiaosu (IFAJ). Meanwhile, another fermentation was carried out by following the preparation procedure of IFAJ but without the inoculation of *Plantilactobacillus plantarum*; the result was nominated as spontaneously fermented *Akebia trifoliata* fruit Jiaosu (SFAJ). SFAJ and IFAJ were prepared in triplicate and samples were collected every 6 days during the fermentation period. The samples were centrifuged (at 10,000 rpm for 15 min), and the supernatant was stored at $-80$ °C for subsequent analysis.

### 2.4. Determination of PH and Total Acid Content

The pH was measured with a high-accuracy pH meter (FE28-Standard, METTLER TOLEDO, Zurich, Switzerland). The total acid content was measured with a potentiometric titrator (916 Ti-Touch, Metrohm, Herisau, Switzerland) and the result was converted into grams of lactic acid equivalent per 100 mL of the liquid sample (g/100 mL). The total acid content was calculated using Equation (1):

$$\text{Total acid content}/(\text{g}/100 \text{ mL}) = \frac{[c \times (V_1 - V_2)] \times k \times F}{V} \times 100 \tag{1}$$

where c is the concentration of the sodium hydroxide standard titration solution (mol/L), $V_1$ is the consumption volume of sodium hydroxide standard titration solution when titrating the sample (mL), $V_2$ is the consumption volume of sodium hydroxide standard titration solution when conducting the blank experiment (mL), k is the conversion coefficient of lactic acid (0.090 kg/mol), F is the dilution ratio of sample, and V is the volume of the sample (mL).

## 2.5. Determination of Bioactive Components

### 2.5.1. Determination of Phenol Content

The determination of phenol content was carried out based on the Folin-Ciocalteu method [20], with some modifications. In brief, 0.1 mL of the sample was diluted with 0.4 mL of deionized water and mixed with Folin-Ciocalteu reagent (10%, $v/v$, 2.5 mL). The reaction process lasted for 3 min prior to adding the $Na_2CO_3$ solution (7.5%, $w/v$, 2.0 mL). The mixture was incubated at 25 °C in the dark for 30 min before measuring the absorbance at 765 nm. The standard curve was obtained with gallic acid.

### 2.5.2. Determination of Flavonoid Content

The flavonoid content was measured, based on the colorimetric method reported by Wolfe et al. [21], with some modifications. In summary, 0.4 mL of the sample was left to react with $NaNO_2$ solution (5%, $w/v$, 0.15 mL) for 6 min. Then, $Al(NO_3)_3$ solution (10%, $w/v$, 0.15 mL) was added and left to react for 6 min. Next, NaOH solution (4%, $w/v$, 2.0 mL) was added. After 15 min, the mixture was diluted with deionized water to 5.0 mL, followed by obtaining the absorbance at 510 nm. The standard curve was made with rutin.

### 2.5.3. Determination of Terpenoid Content

The terpenoid content was determined, based on the colorimetric method reported by Fang et al. [22], with slight modifications. Briefly, 3 µL of the sample was mixed with $H_2SO_4$-methanol (6/1, $v/v$, 2 mL) solution and incubated at 60 °C for 10 min, followed by cooling to room temperature before obtaining the absorbance at 410 nm. The standard curve was created with oleanane.

## 2.6. Antioxidant Activity

### 2.6.1. DPPH Radical Scavenging Test

The DPPH determination was performed, based on the colorimetric method adopted by Blois [23] with some modifications. Briefly, the sample (0.2 mL) was diluted 10-fold with deionized water, then DPPH solution (0.1 mmol/L, dissolved in methanol, 4.0 mL) and Tris-HCl buffer solution (50 mmol/L, pH = 7.4, 0.45 mL) were added, followed by incubation at 25 °C for 30 min before recording the absorbance at 517 nm. The scavenging activity was calculated using Equation (2):

$$\text{DPPH radical scavenging activity}/\% = [1 - \frac{A_1 - A_2}{A_0 - A_2}] \times 100\% \tag{2}$$

where $A_0$ is the absorbance of methanol + DPPH solution, $A_1$ is the absorbance of the sample + DPPH solution, and $A_2$ is the absorbance of the sample + methanol.

### 2.6.2. ABTS Radical Scavenging Test

The ABTS determination was conducted, based on the method used by Re et al. [24] with slight modifications. First, ABTS solution (7.0 mmol/L) was freshly prepared and diluted with phosphate buffer solution (PBS, 5 mmol/L, pH = 7.4) until the initial absorbance at 734 nm reached $0.700 \pm 0.05$. The sample was diluted 50-fold in PBS. The diluted sample (60 µL) was then mixed with ABTS solution (5.0 mL), incubated at 30 °C for 1 h, and cooled to room temperature, before reading the absorbance at 734 nm. The scavenging activity was calculated using Equation (3):

$$\text{ABTS radical scavenging activity}/\% = [1 - \frac{A_1 - A_2}{A_0 - A_2}] \times 100\% \tag{3}$$

where $A_0$ is the absorbance of PBS + ABTS solution, $A_1$ is the absorbance of the sample + ABTS solution, and $A_2$ is the absorbance of the sample + PBS.

2.6.3. Hydroxyl Radical Scavenging Test

The hydroxyl determination was conducted using the colorimetric method employed by Liu et al. [25] with some modifications. Briefly, the sample (0.1 mL) was diluted with deionized water (0.9 mL), followed by the addition of $H_2O_2$ solution (6 mmol/L, 0.7 mL), sodium salicylate solution (20 mmol/L, 0.3 mL), and ferrous sulfate solution (1.5 mmol, 1.0 mL). The mixture was incubated at 37 °C for 1 h and then cooled to room temperature, followed by an absorbance reading taken at 510 nm. The scavenging activity was calculated using Equation (4):

$$\text{Hydroxyl radical scavenging activity}/\% = [1 - \frac{A_1 - A_2}{A_0 - A_2}] \times 100\% \tag{4}$$

where $A_0$ is the absorbance without a sample, $A_1$ is the absorbance of the sample, and $A_2$ is the absorbance without sodium salicylate solution.

*2.7. Network Pharmacology and Molecular Docking Analysis*

2.7.1. Acquisition and Analysis of Target Genes of Active Components and Oxidative Stress

Data on the active components were retrieved from the traditional Chinese medicine systems pharmacology database (TCMSP, https://tcmsp-e.com/tcmspsearch.php, accessed on 15 November 2022). Data on the oxidative stress-related target genes were retrieved from the GeneCards database (https://www.genecards.org/, accessed on 15 November 2022). To filter the intersecting target genes, active components with target genes acting against oxidative stress-related target genes were mapped in a Venn diagram drawn using an online platform (https://bioinfogp.cnb.csic.es/tools/venny/, accessed on 15 November 2022).

2.7.2. Construction of PPI Network

The intersection target genes of active components and oxidative stress were uploaded onto the STRING online platform (https://cn.string-db.org/, accessed on 16 November 2022) to construct a protein–protein interaction (PPI) network, then the Cytoscape 3.7.2 software was used to analyze and visualize the PPI network results.

2.7.3. GO and KEGG Pathway Enrichment Analysis

Gene ontology (GO) analysis and the Kyoto Encyclopedia of Genes and Genomes (KEGG)-based pathway enrichment analysis were performed on the DAVID online platform (https://david.ncifcrf.gov/, accessed on 16 November 2022), on which biological processes (BP), cellular component (CC), molecular function (MF), and the KEGG pathway were analyzed and enriched. The results were visualized on an online platform (http://www.bioinformatics.com.cn/, accessed on 16 November 2022).

2.7.4. Molecular Docking

The 3D structures of the active components and key target proteins were downloaded from the TCMSP database and the Protein Data Bank (PDB) database (https://www.rcsb.org/, accessed on 16 November 2022), respectively. Autodock 4.2.6 software (Scripps Research, La Jolla, CA, USA) was used for the pretreatments of compounds and molecular docking. Finally, the molecular docking results were visualized using PyMOL 2.4.1 software (DeLano Scientific LLC, San Carlos, CA, USA).

*2.8. Protective Effect of SFAJ and IFAJ on Oxidative Stress in HepG2 Cells*

2.8.1. Cell Culture

HepG2 cells were cultured in MEM, containing 1% penicillin-streptomycin solution and 10% heat-inactivated FBS, in an incubator at 37 °C with 5% $CO_2$. When the cell fusion rate reached 80–90%, the cells were passaged and used for experiments after 3 passages.

2.8.2. Cell Viability

The SFAJ and IFAJ were filtered (0.22 μm) and diluted in MEM by 10, 20, 40, 80, and 160 times, obtaining sample solutions at concentrations of 100, 50, 25, 12.5, and 6.25 μL/mL, respectively. The HepG2 cells were seeded on 96-well microplates ($1 \times 10^5$ cells/mL, 100 μL/well) and treated with sample solutions at different concentrations (100 μL/well) for 24-hour incubation. After incubation, the medium was removed, and the cells were washed with PBS (pH = 7.4). The CCK-8 solution (100 μL/mL in PBS) was added (100 μL/well) and incubated at 37 °C for 1 h, followed by an absorbance reading taken at 450 nm. Cell viability was calculated using Equation (5):

$$\text{Cell viability} / \% = \frac{A_{450 \text{ treated}} - A_{450 \text{ blank}}}{A_{450 \text{ control}} - A_{450 \text{ blank}}} \times 100\% \tag{5}$$

where $A_{450 \text{ treated}}$ is the absorbance of wells with treated cells, $A_{450 \text{ control}}$ is the absorbance of wells with untreated cells, and $A_{450 \text{ blank}}$ is the absorbance of wells without cells.

2.8.3. Establishment of an Oxidative Stress Model

HepG2 cells were seeded on 96-well microplates ($1 \times 10^5$ cells/mL, 100 μL/well) and treated with $H_2O_2$ solution at various concentrations (0.1, 0.2, 0.3, 0.4, 0.5, 0.6, and 0.8 mmol/L, 100 μL/well) for a 2-hour incubation. After that, the medium was removed, and the cells were washed with PBS (pH = 7.4). The CCK-8 solution (100 μL/mL in PBS) was added (100 μL/well) and incubated at 37 °C for 1 h, followed by absorbance determination, performed at 450 nm. The cell viability rate was calculated using Equation (4).

2.8.4. Effects of SFAJ and IFAJ on the Cell Viability of $H_2O_2$-Induced HepG2 Cells

The HepG2 cells were seeded on 96-well microplates ($1 \times 10^5$ cells/mL, 100 μL) and treated with sample solutions (the concentration was determined by the result of Section 2.8.2 at 100 μL/well) for the 24-hour incubation period. Thereafter, the medium was removed, and cells were treated with $H_2O_2$ solution (the concentration was determined by the result of Section 2.8.3, at 100 μL/well), and incubated for 2 h. The CCK-8 solution (100 μL/mL in PBS) was added (100 μL/well) and incubated at 37 °C for 1 h, followed by an absorbance measurement taken at 450 nm. The cell viability rate was calculated using Equation (4).

2.8.5. Effects of SFAJ and IFAJ on Intracellular ROS Levels and the Activities of Antioxidant Enzymes in $H_2O_2$-Induced HepG2 Cells

The HepG2 cells were seeded on 24-well microplates ($1 \times 10^5$ cells/mL, 1 mL/well) and treated with sample solutions (the concentration was determined by the result of Section 2.8.2 at 1 mL/well) with 24-hour incubation. After that, the medium was removed and cells were treated with $H_2O_2$ solution (the concentration was determined by the result of Section 2.8.3, 1 mL/well), and incubated for 2 h, followed by removal of the medium and treatment with DCFH-DA (10 μmol/L, 1 mL/well) in a 30-minute incubation. The cells were washed 3 times with MEM (FBS-free) and PBS was added (1 mL/well). The absorbance was measured using a fluorescence microplate reader at an excitation wavelength of 500 nm and an emission wavelength of 525 nm. The ROS level was calculated using Equation (6):

$$\text{ROS level} / \% = \frac{A_{525 \text{ treated}} - A_{525 \text{ blank}}}{A_{525 \text{ control}} - A_{525 \text{ blank}}} \times 100\% \tag{6}$$

where $A_{525 \text{ treated}}$ is the absorbance of wells with treated cells, $A_{525 \text{ control}}$ is the absorbance of wells with untreated cells, and $A_{525 \text{ blank}}$ is the absorbance of wells without cells.

The HepG2 cells were seeded on 6-well microplates ($1 \times 10^5$ cells/mL, 2.5 mL) and treated with sample solutions (the concentration was determined by the result of Section 2.8.2, at 2.5 mL/well) with 24-hour incubation. Then, the medium was removed, and cells were treated with $H_2O_2$ solution (the concentration was determined by the result of Section 2.8.3,

at 2.5 mL/well) and incubated for 2 h. Thereafter, the medium was removed, and RIPA lysis buffer was added (250 μL/well). After complete lysis oscillation, a centrifugation treatment was carried out (15,000 rpm, 10 min, 4 °C), and the supernatant was collected for the determination of the protein contents and activities of antioxidant enzymes (SOD, CAT, and GSH-Px), which were performed following the manufacturer's instructions, using the corresponding kits.

### 2.9. Statistical Analysis

All experiments were performed in triplicate and the data were expressed as means ± standard deviation. The SPSS 26.0 software (SPSS, Chicago, IL, USA) was used for statistical analysis. Experimental data were subject to a one-way analysis of variance (ANOVA), followed by post hoc analysis with Duncan's multiple range tests, and $p < 0.05$ was defined as significantly different. Figures were created using the Origin 2023 software (OriginLab Corp., Northampton, MA, USA).

## 3. Results and Discussion

### 3.1. Changes in PH and Total Acid Content during Fermentation

PH and total acid content are a pair of important indicators that reflect the fermentation status and the metabolism of microorganisms. As seen in Figure 1a, the pH of the SFAJ decreased quickly in the initial 24 days (from 5.33 ± 0.09 to 4.61 ± 0.01) and stabilized around 4.60, reaching 4.59 ± 0.02 by the end of fermentation. Similarly, for IFAJ, the pH dropped significantly during the first 24 days (from 5.30 ± 0.19 to 4.30 ± 0.01) and afterward stabilized in the same way, with a pH of 4.17 ± 0.03 on day 72. As shown in Figure 1b, the total acid content of SFAJ rose rapidly ($p < 0.05$) in the first 42 days and stabilized until the end of testing (1.78 ± 0.05 on day 72). Meanwhile, the total acid content of IFAJ presented a significant increase in the first 42 days ($p < 0.05$), and a stable trend was also obtained in the late period of fermentation. The phenomena mentioned above could be explained as follows. In the early stages of fermentation for both SFAJ and IFAJ, the predominant microorganisms, such as *Plantilactobacillus plantarum*, converted a portion of the organic substances (such as ethanol and hexose) into acids (lactic acid, acetic acid, citric acid, etc.) or other acidic compounds [26,27], resulting in a total acid content increase and a decline in pH. In the middle and late stages, the respiration of the microorganisms turned weak due to the decrease in nutrients, which directly led to a deceleration in the formation rate of acids. Besides this, the results also suggested a more thorough fermentation of IFAJ, which was reflected by its eventual higher acid content and lower pH than SFAJ.

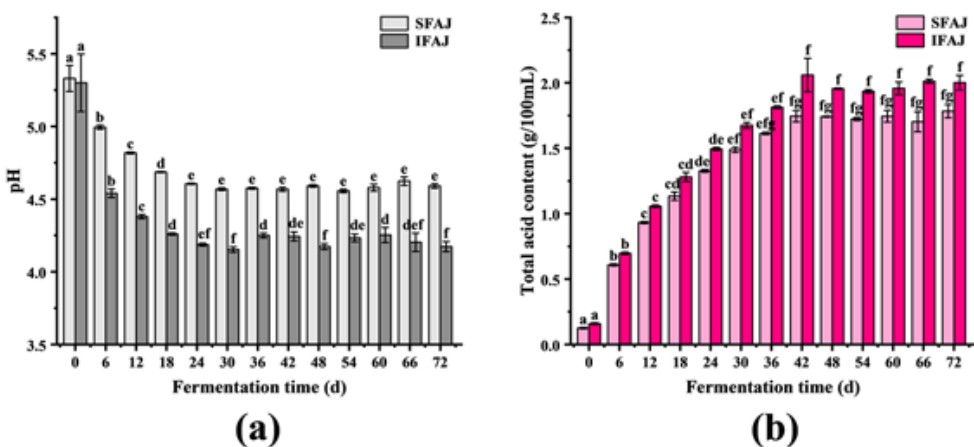

**Figure 1.** Changes in pH (**a**) and total acid content (**b**) during the fermentation of SFAJ and IFAJ. Different letters indicate significant differences in the same group among samples from different fermentation time points ($p < 0.05$).

### 3.2. Changes in Bioactive Components during Fermentation

3.2.1. Changes in Phenol Content during Fermentation

Their unique physiological functions, such as inhibiting the formation of free radicals, enhancing antioxidant defense mechanisms in cells, and impairing the action of pro-oxidative enzymes, make phenolic compounds applicable in health care, processed foods, cosmetics industries, auxiliary medicine remedies, etc. [28] As shown in Figure 2a, the phenol content of SFAJ increased significantly ($p < 0.05$) in the first 24 days (from $0.58 \pm 0.04$ mg/mL to $1.08 \pm 0.04$ mg/mL). Afterward, it dropped to $0.98 \pm 0.01$ mg/mL on day 36 and rose again to $1.15 \pm 0.04$ mg/mL on day 48, then stabilized ($p > 0.05$) until it reached $1.18 \pm 0.02$ mg/mL on the final day. The phenol content of IFAJ increased significantly (from $0.60 \pm 0.02$ mg/mL to $1.16 \pm 0.05$ mg/mL) during the first 24 days of fermentation, with an increase of 93.33% compared with day 0 ($p < 0.05$). From day 24 to day 36, the phenol content slightly decreased. However, it rose again in the following 12 days (to $1.19 \pm 0.07$ mg/mL) and no significant difference was observed after day 42 ($p > 0.05$), with the maximum ($1.26 \pm 0.02$ mg/mL) on day 72.

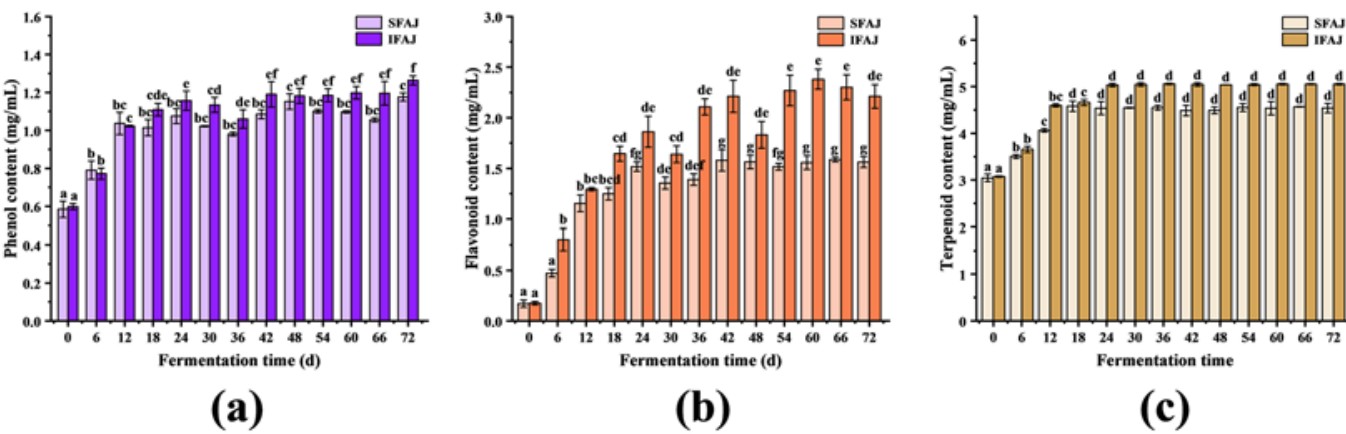

**Figure 2.** Changes in phenol content (**a**), flavonoid content (**b**), and terpenoid content (**c**) during the fermentation of SFAJ and IFAJ, respectively. Different letters indicate significant differences in the same group among samples from different fermentation time points ($p < 0.05$).

It was worth noting that the changing trends of SFAJ and IFAJ were extremely similar. In the early stages of fermentation, due to the high osmotic pressure formed by the high sugar content in the broth, the phenolic substances in *Akebia trifoliata* fruit might be released and dissolved into the culture broth in quantity. Besides this, secondary metabolites such as organic acids, which are produced during the growth and reproduction of microorganisms, could also promote the dissolution process [29], thus resulting in a rapid increase in phenol content. During the middle stages, since phenols at high concentrations (such as chlorogenic acid) were bacteriostatic, certain microorganisms started to decompose part of the phenols in order to maintain their normal growth and reproduction [30], which led to a minor decrease in the phenol concentrations. In the late stages, probably due to the degradation of macromolecular phenols into micromolecular phenols by microorganisms that are resistant to a high-concentration phenolic environment [31], the phenol content continued to increase. Generally, the phenol content of IFAJ was higher than that of SFAJ in the middle and late stages of fermentation, which indicated that inoculation fermentation by *Plantilactobacillus plantarum* improved the generation of antioxidant phenol metabolites; on the other hand, inoculated bacteria have better tolerance of phenols than spontaneously fermented microorganisms.

3.2.2. Changes in Flavonoid Content during Fermentation

Flavonoids have multiple functions, such as activating antioxidant enzymes, suppressing prooxidative enzymes as antioxidants, preventing the progression of cancer,

inhibiting chronic inflammation, etc. [32] As exhibited in Figure 2b, the flavonoid content of SFAJ slowly increased during the first 6 days of fermentation ($p > 0.05$) and rose rapidly in the following 6 days ($p < 0.05$); it increased with fluctuations, eventually reaching $1.56 \pm 0.05$ mg/mL. In the case of IFAJ, the flavonoid content increased significantly during the first 24 days of fermentation (from $0.18 \pm 0.02$ mg/mL to $1.86 \pm 0.15$ mg/mL) ($p < 0.05$). In the following days, the flavonoid content fluctuated frequently but showed an overall upward trend, finally reaching $2.21 \pm 0.32$ mg/mL on day 72. In comparison, the flavonoid antioxidant substances value produced by *Plantilactobacillus plantarum* bacteria inoculation fermentation for IFAJ was 41.67% higher than those produced by spontaneous fermentation for SFAJ, which further reflected the advantages of artificially inoculated fermentation.

The initial decrease in flavonoid content might be attributed to the rapid growth, reproduction, and vigorous metabolism of microorganisms, which needed the flavonoids as another carbon source to consume apart from saccharides. In addition, the high osmotic pressure in the fermentation broth, coupled with the life activities of microorganisms, caused the plant cells to rupture and the flavonoids to leak, which directly led to the rapid rise in flavonoid substances. Therefore, the flavonoid content was in a relatively dynamic state of change, depending on consumption and generation. In the middle and late periods of fermentation, the flavonoids originating from the seeds dissolved into the broth, due to the increase in ethanol concentration [29] (the detailed data for ethanol content are shown in Figure S1 in the Supplementary Materials). In addition, with the assistance of specific enzymes, a portion of other carbon sources could be converted into compounds with the structure of pyrocatechol during microbial metabolism [33], which also contributed to the increase in flavonoid content as determined by colorimetry.

### 3.2.3. Changes in Terpenoid Content during Fermentation

Terpenoids, as some of the most diverse natural products, have been used in medicine and industry for decades [34]. As shown in Figure 2c, the terpenoid content of SFAJ continuously increased in the first 18 days ($p < 0.05$) and remained stable at around 4.50 mg/mL, eventually being recorded at $4.54 \pm 0.10$ mg/mL on day 72. The terpenoid content of IFAJ presented an obviously increasing trend in the first 24 days of fermentation (from $3.07 \pm 0.02$ mg/mL to $5.03 \pm 0.03$ mg/mL) ($p < 0.05$). Afterward, the terpenoid content stabilized and reached $5.05 \pm 0.01$ mg/mL on day 72, over 1.1 times that in SFAJ. This was consistent with the formation trend of phenol and flavonoid metabolites during the EPJ fermentation, in which the artificially inoculated fermentation was endowed with more abundant antioxidant metabolites. Notably, the terpenoid content obtained a relatively high value in the early stages of fermentation in both groups, which was attributed to multiple factors (such as high osmotic pressure, the life activities of microorganisms, conversion from or formation by other components [35,36], synergistic effects under complex conditions, etc.) and was of great significance to the optimization of IFAJ fermentation time.

### *3.3. Changes in Antioxidant Activities during Fermentation*
### 3.3.1. Changes in DPPH Radical Scavenging Activity during Fermentation

The DPPH free radical scavenging test is a simple and rapid method that is commonly used to evaluate the antioxidant activity of antioxidants [37]. As shown in Figure 3a, the DPPH free radical scavenging activity of SFAJ increased rapidly from $42.09 \pm 1.22\%$ on day 0 to $82.66 \pm 1.30\%$ on day 12 ($p < 0.05$), and stabilized at around 84%, finally reaching $84.27 \pm 0.18\%$ on day 72. In the case of IFAJ, the DPPH free radical scavenging activity significantly rose from $53.04 \pm 1.23\%$ on day 0 to $93.61 \pm 1.30\%$ on day 12 ($p < 0.05$), with an increase of 76.49%, and remained stable at around 90% until reaching $94.00 \pm 0.78\%$ at the end of fermentation. The trends of DPPH radical scavenging ability in both groups during fermentation were similar to the results reported by Jiang et al., in which the DPPH free radical scavenging activity of grape Jiaosu sharply increased to 88.15% in the first 24 days and then stabilized [38]. For the fermentation culture broth, using the same fermentation time and the same dosage, the artificially inoculated fermentation product of

IFAJ showed better DPPH radical scavenging activity than the spontaneous fermentation product of SFAJ, which finding was speculated to be related to the higher contents of phenols, flavonoids, and terpenoids in the artificially inoculated fermentation product of IFAJ. According to Shahidi et al., the presence of phenols substantially contributed to the scavenging of DPPH radicals by virtue of their excellent hydrogen-donating ability [39].

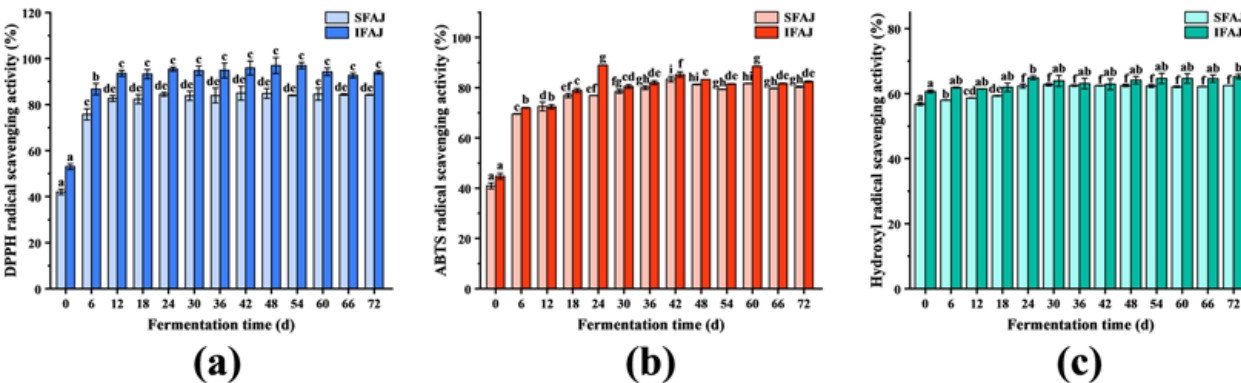

**Figure 3.** Changes in DPPH (**a**), ABTS (**b**), and hydroxyl (**c**) radical scavenging activity during the fermentation of SFAJ and IFAJ, respectively. Different letters indicate significant differences in the same group among samples from different fermentation time points ($p < 0.05$).

### 3.3.2. Changes in ABTS Radical Scavenging Activity during Fermentation

The ABTS radical scavenging test is another widely accepted and extensively used antioxidant assay [40]. As shown in Figure 3b, the ABTS radical scavenging activity of SFAJ increased rapidly during the first 6 days of fermentation ($p < 0.05$) and gradually stabilized with an upward trend until it reached $80.42 \pm 0.22\%$ at the end of fermentation. The scavenging activity of IFAJ increased significantly in the initial 24 days (from $44.80 \pm 1.14\%$ to $88.98 \pm 0.05\%$) ($p < 0.05$) and slightly decreased in the following 6 days, then stabilized with minor fluctuations until it reached $82.43 \pm 0.21\%$ on day 72, with an increase of 84.00% compared with that on day 0. The trend of ABTS radical scavenging ability was similar to the results reported by Wang et al., in which the ABTS radical scavenging activity of kidney bean Jiaosu experienced a quick rise in the first 32 h and then reached a stable state [29]. On the one hand, fermentation enhanced the antioxidant capacity of *Akebia trifoliata* fruit since the higher the concentration of phenols and flavonoids in the culture broth, the stronger the ability to scavenge ABTS free radicals, although the scavenging ability also depended on the molecular weight, the number of aromatic rings, the nature of the hydroxyl substituent, etc. [41,42]. On the other hand, compared with spontaneous fermentation, the product of IFAJ produced by the artificial inoculation of *Plantilactobacillus plantarum* possessed better ABTS radical inhibition activity.

### 3.3.3. Changes in Hydroxyl Radical Scavenging Activity during Fermentation

The hydroxyl radical, which is recognized as one of the most reactive free radicals formed in biological systems, is considered highly destructive in free radical pathology because of the prevalence of the damage it causes in living cells [43]. As shown in Figure 3c, the hydroxyl radical scavenging activity of SFAJ increased in the initial 24 days (from $56.82 \pm 0.44\%$ to $62.34 \pm 0.60\%$) ($p < 0.05$), followed by a stable period until the end of testing ($62.48 \pm 0.01\%$ on day 72). The hydroxyl radical scavenging activity of IFAJ experienced a slow and slight increase during the fermentation process, reaching $65.22 \pm 0.71\%$ on day 72, with a total increase of 7.46% compared with that on day 0. In comparison with the DPPH and ABTS radical scavenging activity, the hydroxyl radical scavenging activity of SFAJ and IFAJ was much less enhanced during the whole fermentation period. In spite of this, the experimental group inoculated by *Plantilactobacillus plantarum* had stronger hydroxyl radical scavenging activity than the spontaneous fermentation group, which still confirmed the advantages of inoculation fermentation.

### 3.4. Correlation Heatmap and Principal Component Analysis of Metabolites and Antioxidant Activities

In order to explore the relationship between the fermentation metabolites of *Akebia trifoliata* fruit and its enhanced antioxidant activity, the correlation between the metabolites and various antioxidant indexes was analyzed. The correlation heatmap analyses of metabolites and antioxidant activity are shown in Figure 4a,c. The contents of phenol, flavonoids, and terpenoids were positively correlated with DPPH, ABTS, and hydroxyl radical scavenging activities, respectively, with correlation coefficients greater than 0.59 ($p < 0.05$). This is consistent with previous reports that phenols, flavonoids, and terpenoids possessing antioxidant capacities showed strong free radical scavenging activities [44,45]. Besides this, a strong positive correlation was observed between the total acid content and DPPH, ABTS, and hydroxyl radical scavenging activity, with correlation coefficients of above 0.79, which was probably related to the organic acids with antioxidant properties (such as ascorbic acid, citric acid, etc. [46]) that were produced and accumulated during the fermentation process. Correspondingly, pH was negatively correlated with the DPPH, ABTS, and hydroxyl radical scavenging activities. The preliminary results indicated that phenols, flavonoids, terpenoids, and organic acids that were metabolized during the fermentation of *Akebia trifoliata* fruit contributed to the enhancement of the antioxidant activity of the fermentation broth.

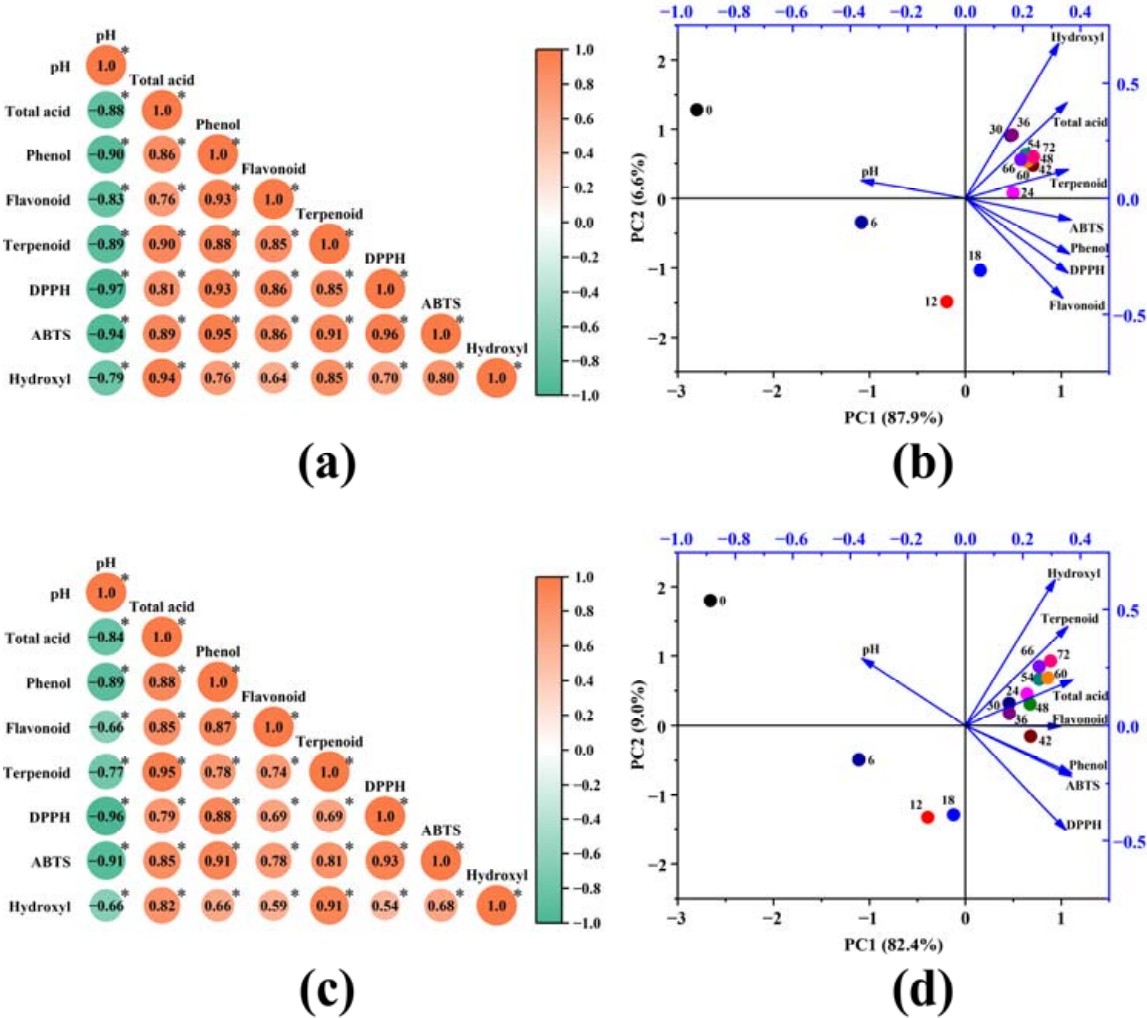

**Figure 4.** Correlation heatmap and principal component analysis of the metabolites and antioxidant activity of SFAJ (**a**,**b**) and IFAJ (**c**,**d**) during fermentation, respectively. In (**a**,**c**), the symbol * indicates $p < 0.05$; in (**b**,**d**), the dots represent samples at different fermentation time points, and arrows represent the loads of the principal components.

In the fermentation process of *Akebia trifoliata* fruit, four active ingredients (acids, phenols, flavonoids, and terpenoids), three antioxidant indexes (DPPH, ABTS, and hydroxyl radical scavenging activity), and pH value could characterize the quality of the fermentation product. At the same time, using these eight indicators to characterize the quality of *Akebia trifoliata* fruit fermentation products would bring a great deal of extra work to the commercial production of and research into *Akebia trifoliata* fruit. Therefore, it is necessary to establish a comprehensive evaluation index for *Akebia trifoliata* fruit fermentation products. Considering the strong correlation between multiple indicators, principal component analysis (PCA) was used to reduce the dimensionality of high-dimensional data during the fermentation of SFAJ and IFAJ. In the PCA of SFAJ (Figure 4b), the cumulative contribution of principal component 1 (PC1) and principal component 2 (PC2) was 94.5%, with eigenvalues of 7.03 and 0.52, respectively, while for IFAJ (Figure 4d), the cumulative contribution of (PC1) and (PC2) reached 91.4%, with eigenvalues of 6.59 and 0.72, respectively. These findings indicate that this is a reliable way to reflect most of the changes in indexes with PC1 and PC2 for SFAJ and IFAJ. With the prolongation of fermentation time, the sample points started to move from the second quadrant to the third quadrant and finally aggregated in the first quadrant (after day 24), then gradually stayed farther away from the coordinate origin, which meant that samples in the middle and late stages of fermentation shared more in common in composition and antioxidant activity and gained more of a contribution from PC1 and PC2. The variables (except pH) were distributed in the first and fourth quadrants and were all close to the sample points on day 24, manifesting their close relationship, which was consistent with the results of the correlation heatmap analysis. The establishment of a comprehensive evaluation index (CEI) based on PCA analysis was conducive to the judgment of fermentation time and the accurate control of fermentation. The CEI for SFAJ and IFAJ was calculated using Equation (7):

$$\text{CEI} = \frac{\text{Eig1} \times \text{FAC1} + \text{Eig2} \times \text{FAC2}}{\text{Eig1} + \text{Eig2}} \tag{7}$$

where Eig1 and Eig2 are the eigenvalues of PC1 and PC2, and FAC1 and FAC2 are the factor scores of PC1 and PC2.

The CEI for SFAJ and IFAJ is summarized in Table 1. With the extension of fermentation time, the CEI value for both SFAJ and IFAJ increased linearly in the first 9 days of fermentation, the linear increase rate slowed down in the 9th to 24th days of fermentation, and the CEI value was basically maintained at a relatively stable level on day 42, indicating that the fermentation might have good quality on day 42, which was basically consistent with the aforementioned results of chemical antioxidant activity. Without taking the time cost into account, the sample on day 72 doubtless showed the best fermentation properties in terms of antioxidant activity.

**Table 1.** Comprehensive evaluation index (CEI) of SFAJ and IFAJ during fermentation.

| Fermentation Time (d) | SFAJ | | | IFAJ | | |
|:---:|:---:|:---:|:---:|:---:|:---:|:---:|
| | FAC1 | FAC2 | CEI | FAC1 | FAC2 | CEI |
| 0 | −2.80 | 1.28 | −2.52 | −2.66 | 1.80 | −2.22 |
| 6 | −1.09 | −0.35 | −1.04 | −1.11 | −0.49 | −1.05 |
| 12 | −0.20 | −1.49 | −0.29 | −0.39 | −1.33 | −0.48 |
| 18 | 0.15 | −1.04 | 0.07 | −0.12 | −1.29 | −0.24 |
| 24 | 0.49 | 0.07 | 0.46 | 0.64 | 0.45 | 0.62 |
| 30 | 0.47 | 0.90 | 0.50 | 0.46 | 0.32 | 0.44 |
| 36 | 0.49 | 0.91 | 0.52 | 0.46 | 0.17 | 0.43 |
| 42 | 0.70 | 0.47 | 0.68 | 0.68 | −0.16 | 0.60 |
| 48 | 0.68 | 0.55 | 0.67 | 0.68 | 0.31 | 0.64 |
| 54 | 0.63 | 0.62 | 0.63 | 0.77 | 0.67 | 0.76 |

**Table 1.** *Cont.*

| Fermentation Time (d) | SFAJ | | | IFAJ | | |
|---|---|---|---|---|---|---|
| | FAC1 | FAC2 | CEI | FAC1 | FAC2 | CEI |
| 60 | 0.64 | 0.52 | 0.63 | 0.86 | 0.68 | 0.84 |
| 66 | 0.58 | 0.55 | 0.57 | 0.77 | 0.85 | 0.78 |
| 72 | 0.71 | 0.59 | 0.70 | 0.89 | 0.93 | 0.89 |

*3.5. Network Pharmacology and Molecular Docking Analysis*

The *Akebia trifoliata* fruit is also known as a traditional Chinese medicine called *Akebiae Frucyus*, which contains active components with unique pharmacological properties that are mainly seen through antioxidation, anti-free radicals, and other physiological functions. Herein, we attempt to elucidate the potential antioxidation mechanism of SFAJ and IFAJ, using network pharmacology analysis and a molecular docking technique, for the purpose of identifying the basis of their pharmacological activities.

3.5.1. Target Genes Analysis of Active Components and Oxidative Stress

Network pharmacology analysis was carried out to further illustrate the mechanism of the action of antioxidation activity of SFAJ and IFAJ. The active components and target genes of the *Akebia trifoliata* fruit are summarized in Table 2; a total of 12 components were collected, among which 7 components were obtained with target genes. Based on the data, an active components target network was created and is shown in Figure 5a, comprising 7 active components (green squares), 40 corresponding target genes (blue circles), and 61 interactions (black edges). The above active components and their corresponding target genes might play critical roles in anti-oxidative stress treatment by SFAJ and IFAJ. β-sitosterol possessed the most target genes, in terms of number, and interacted with 25 target genes. Among the target genes, NCOA2, PTGS1, and PTGS2 interacted with 5, 4, and 4 active components, respectively. The results above suggest that the *Akebia trifoliata* fruit is a medicine characterized by multiple active components, multiple target genes, and complex interactions. As is shown in the Venn diagram below (Figure 5b), a total of 9629 oxidative stress-related target genes were collected and 29 active components/oxidative stress common target genes were filtered.

**Table 2.** Active components and target genes of *Akebia trifoliata* fruit.

| Component ID | Component | Oral Bioavailability (%) | Drug-Likeness | Target Genes |
|---|---|---|---|---|
| MOL010928 | [(2R)-2,3-dihydroxypropyl] octadecanoate | 25.20 | 0.29 | N/A * |
| MOL010929 | glyceryl linolenate | 38.14 | 0.31 | PTGS1, PTGS2 |
| MOL000263 | oleanolic acid | 29.02 | 0.76 | CASP9, CASP3, Hmox1, ICAM1, AMY2 |
| MOL002882 | [(2R)-2,3-dihydroxypropyl] (Z)-octadec-9-enoate | 34.13 | 0.30 | N/A |
| MOL000357 | sitogluside | 20.63 | 0.62 | PGR, PTGS1, Chrm3, Kcnh2, CHRM1, SCN5A, PTGS2, Htr3a, RXRA, Adra1b, ADRB2, Adra1d, NCOA2, CAM |
| MOL000358 | β-sitosterol | 36.91 | 0.75 | PGR, NCOA2, PTGS1, PTGS2, Kcnh2, Chrm3, CHRM1, SCN5A, CHRM4, ADRA1A, CHRM2, Adra1b ADRB2, CHRNA2, SLC6A4, OPRM1, GABRA1, BCL2, BAX, CASP9, CASP3, CASP8, PRKCA, PON1, MAP2 |
| MOL000359 | sitosterol | 36.91 | 0.75 | PGR, NCOA2, NR3C2 |
| MOL000551 | hederagenol | 22.42 | 0.74 | N/A |

**Table 2.** *Cont.*

| Component ID | Component | Oral Bioavailability (%) | Drug-Likeness | Target Genes |
|---|---|---|---|---|
| MOL000069 | palmitic acid | 19.3 | 0.1 | CTSD, ADH1C, PTGS1, PTGS2, RHO, NCOA2, BCL2, Il10, TNF, SLC22A5, PCYT1A, NCOA2 |
| MOL007254 | arjunolic acid | 23.22 | 0.72 | N/A |
| MOL008121 | 2-mono-olein | 34.23 | 0.29 | NCOA2 |
| MOL008218 | 1-mono-olein | 34.13 | 0.30 | N/A |

* N/A indicates that the target genes are not available.

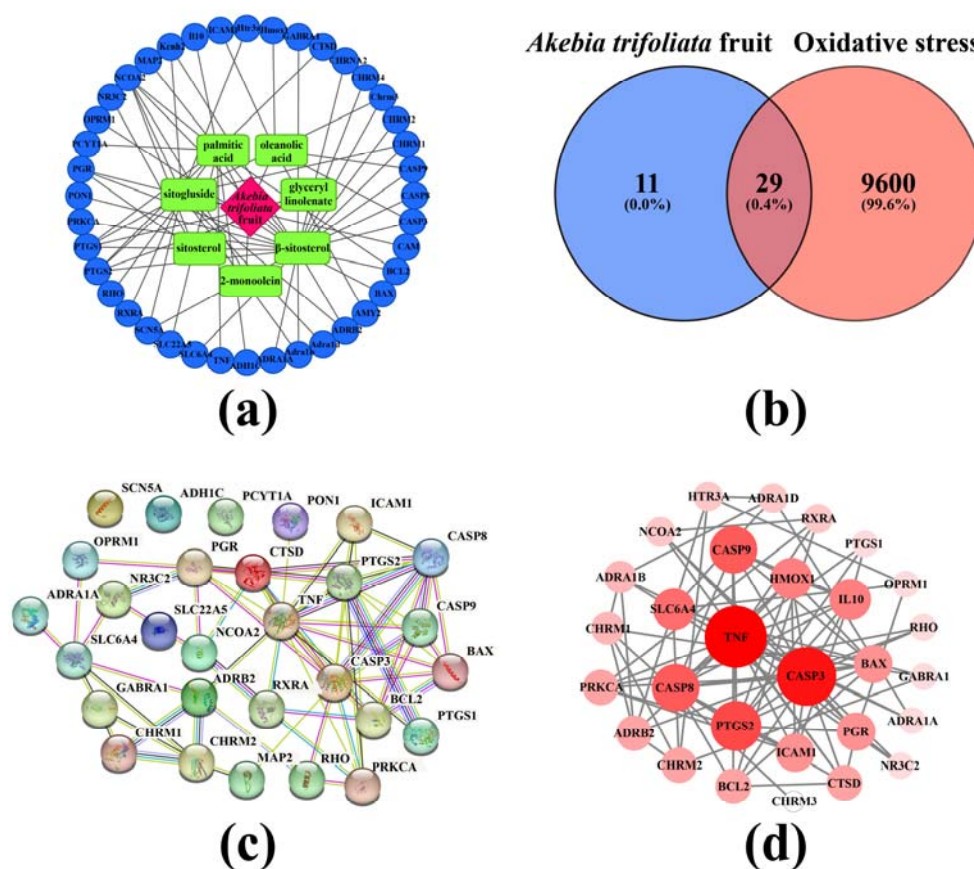

**Figure 5.** Active components and targets network of *Akebia trifoliata* fruit (**a**), Venn diagram of target active components and oxidative stress (**b**), the protein–protein interaction network (PPI) (**c**), and high interaction modules based on the active component–oxidative stress intersection target genes (**d**). In (**b**), the figure and percentage in the blue area represent the number of active component-related target genes of *Akebia trifoliata* fruit and their proportion, the figure and percentage in the red area represent the number of oxidative stress-related target genes and their proportion, and the figure and percentage in the overlapping area represent the number of active components and oxidative stress common target genes and their proportion. In (**d**), node size and color represent the degree of connectivity.

### 3.5.2. Analysis of the PPI Network

The collection of 36 active components–oxidative stress intersection target genes was performed to construct the PPI network, aiming to disclose the relationships among interacting genes and proteins. The PPI network (Figure 5c) consisted of 29 nodes and 64 edges; the average node degree was 4.41 and the enrichment *p*-value was less than $1.0 \times 10^{-16}$. TNF, CASP3, CASP8, PTGS2, and other proteins were highly interactive with each other. Then, the data were imported to Cytoscape 3.7.2 to construct high-interaction modules (Figure 5d). The results showed that the top 5 highly interacted-with target genes were TNF, CASP3, PTGS2, CASP8, and CASP9. However, in view of the oral bioavailability

(≥30%) and drug-likeness (≥0.18) of the active components [47], active components such as glyceryl linolenate and β-sitosterol, and the target genes CASP3, PTGS2, CASP8, and CASP9 were selected for subsequent molecular docking.

### 3.5.3. Analysis of GO and KEGG Pathway Enrichment

To understand the antioxidation mechanisms of action of SFAJ and IFAJ against oxidative stress, functional enrichment analysis was performed on the target genes of bioactive components from *Akebia trifoliata* fruit. Gene ontology (GO) analysis was adopted to evaluate the biological processes (BP), cell components (CC), and molecular functions (MF), and the top 9 BP, 6 CC, and 4 MF catalogs were selected ($p < 0.01$) for visualization (Figure 6a). The results showed that the BP terms were related to the regulation of the apoptotic process, the response to xenobiotic stimulus, the response to DNA damage, and the response to drug- and other BP-related GO terms. The CC terms involved the integral components of membranes, neuron projection, the membrane raft, and the caveola. The MF terms mainly influenced the activities of the alphal-adrenergic receptor, neurotransmitter receptor, G-protein-coupled acetylcholine receptor, and identical protein binding. Therefore, to some extent, this rich GO function could be used to explain the antioxidant and other pharmacological activities of the fermented *Akebia trifoliata* fruit. In addition, KEGG pathway enrichment analysis was performed to excavate further mechanisms. As exhibited in Figure 6b, the top 67 enriched pathways are visualized using a bubble chart ($p < 0.01$). The main pathways of enrichment included apoptosis in multiple species, African trypanosomiasis, legionellosis, platinum drug resistance, and the p53 signaling pathway, etc., suggesting the potential involvements and mechanisms for fermented *Akebia trifoliata* fruit in the inhibition of the oxidation process [48].

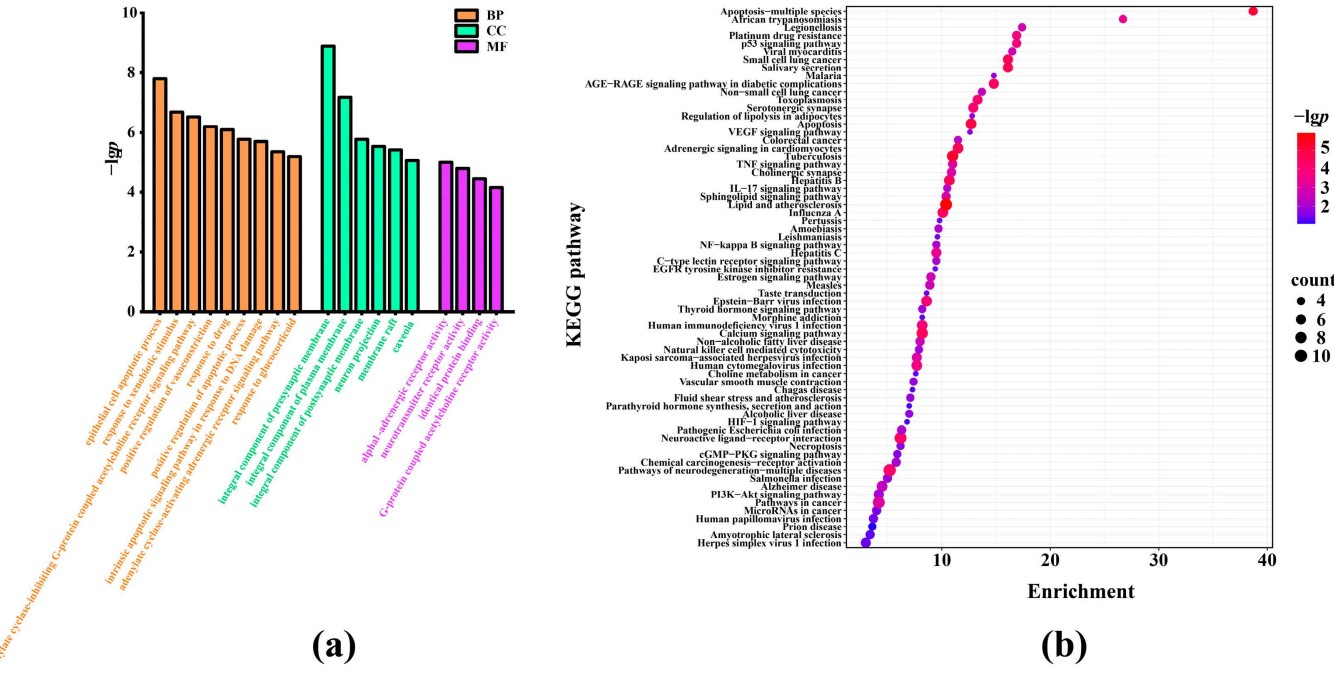

**(a)**　　　　　　　　　　　　　　**(b)**

**Figure 6.** Gene ontology (GO) analysis (**a**) and the Kyoto Encyclopedia of Genes and Genomes (KEGG) pathway enrichment analysis of active components-oxidative stress targets (**b**). In (**a**), BP, CC, and MF represent biological processes, cell components, and molecular functions, respectively; in (**b**), the degree of gene enrichment is represented by the abscissa, the amount of gene enrichment is represented by the bubble size, and the *p*-value is represented by the color depth.

### 3.5.4. Molecular Docking

Based on the network analysis, two active components (glyceryl linolenate and β-sitosterol) and four targets (PTGS2, CASP3, CASP8, and CASP9) were screened by phar-

macology analysis for molecular docking. The simulated docking diagrams and relevant parameters are shown in Figure 7 and Table 3, in which a binding energy of less than −5 kcal·mol$^{-1}$ indicated a strong binding force [49]. Based on this criterion, the binding sites on the four target proteins were all abundant for β-sitosterol to form firm bindings. Among them, the binding energy of β-sitosterol with PTGS2 residues (SER-119, ARG-120, PRO-86, GLU-524, MET-471 and LYS-473) was −6.23 kcal·mol$^{-1}$, indicating strong binding. The molecular docking results also indicated that β-sitosterol was probably the key compound in the pharmacological action of *Akebia trifoliata* fruit, while PTGS2, CASP3, CASP8, and CASP9 might be the key targets in the pharmacological action of fermented *Akebia trifoliata* fruit, which elucidates the antioxidation mechanism of SFAJ and IFAJ. Therefore, in vitro validation should be conducted to verify the results accordingly.

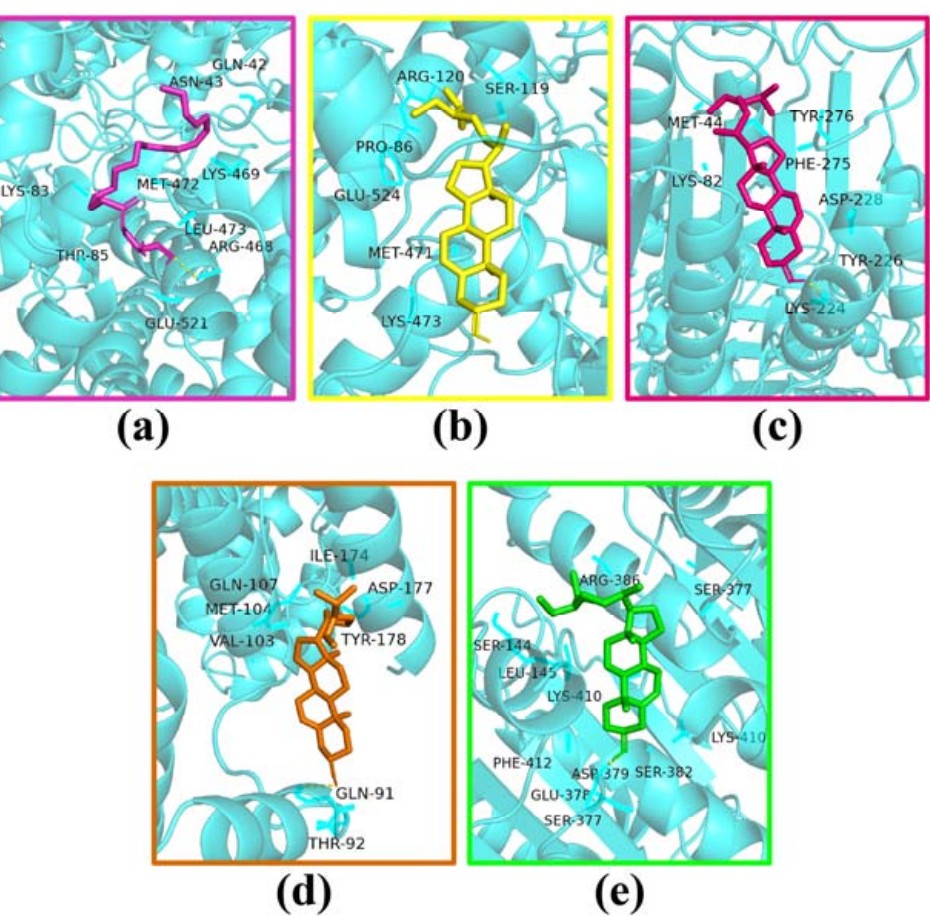

**Figure 7.** Molecular docking of glyceryl linolenate with PTGS2 (**a**), β-sitosterol with PTGS2 (**b**), CASP3 (**c**), CASP8 (**d**), and CASP9 (**e**).

**Table 3.** Results of molecular docking for the active components of *Akebia trifoliata* fruit and target proteins.

| Active Component | Target Genes | PDB ID | Binding Energy (kcal·mol$^{-1}$) | Inhibition Constant (μM) |
|---|---|---|---|---|
| glyceryl linolenate | PTGS2 | 5IKQ | −1.22 | 127.75 |
| β-sitosterol | PTGS2 | 5IKQ | −6.23 | 27.02 |
| β-sitosterol | CASP3 | 7SEO | −5.22 | 148.21 |
| β-sitosterol | CASP8 | 6X8H | −6.19 | 28.88 |
| β-sitosterol | CASP9 | 2AR9 | −5.86 | 50.60 |

### 3.6. Protective Effect on $H_2O_2$-Induced Oxidative Stress Injury in HepG2 Cells

Based on the CEIs of SFAJ and IFAJ during fermentation, samples of SFAJ and IFAJ that had been fermented for 72 days were used as experimental subjects for the study of anti-oxidative stress in vitro, based on the HepG2 cell model. In the following experiments, the groups of untreated, treated by $H_2O_2$ solution, SFAJ, and IFAJ were nominated as the Control, Model, SFAJ, and IFAJ, respectively.

#### 3.6.1. Cell Viability

The viability of HepG2 cells treated with SFAJ and IFAJ at different concentrations is shown in Figure 8a. Compared with the cells in the Control group, SFAJ and IFAJ at high concentrations significantly inhibited the viability of HepG2 cells, which increased correspondingly with an increase in the dilution ratio ($p < 0.05$). Among them, the cell viability of SFAJ that had been diluted 160-fold and IFAJ that had been diluted 80-fold reached $95.17 \pm 3.08\%$ and $95.72 \pm 2.81\%$, respectively, which was not significantly different from the values of the Control group ($p > 0.05$), suggesting that SFAJ and IFAJ at these or lower concentrations exerted no toxicity on the HepG2 cells. Therefore, SFAJ and IFAJ that had been diluted 160-fold in MEM were used as experimental samples for subsequent studies.

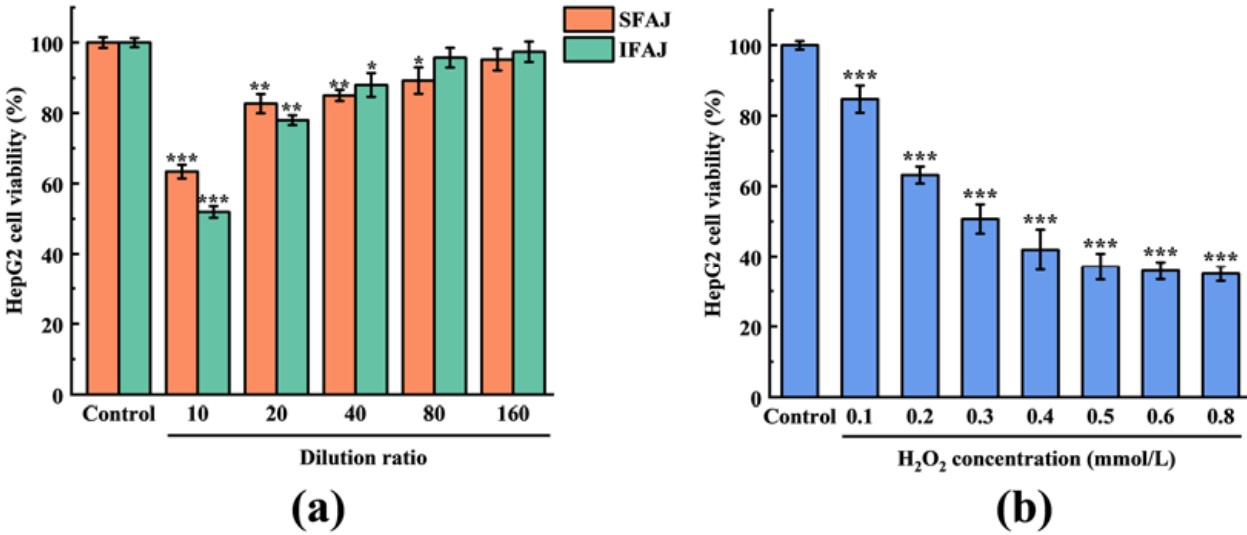

**Figure 8.** Effects of SFAJ and IFAJ at different concentrations on the viability of HepG2 cells (**a**) and the effects of $H_2O_2$ solution at different concentrations on the viability of HepG2 cells (**b**). Compared with the Control group, the symbols *, **, and *** indicate $p < 0.05$, $p < 0.01$, and $p < 0.001$, respectively.

#### 3.6.2. Establishment of the Oxidative Stress Model

$H_2O_2$ can penetrate biological membranes and generate ROS molecules (such as hydroxyl radicals) as intermediates in cells, causing serious damage to biological macromolecules such as nucleic acids and proteins [50]. Therefore, $H_2O_2$ was used as a stimulant to establish an oxidative stress model. The viability of HepG2 cells treated with $H_2O_2$ solution at different concentrations is presented in Figure 8b. When the concentration of the $H_2O_2$ solution was in the range of 0.1~0.8 mmol/L, with an increase in concentration, the cell proliferation inhibition was significant, compared with the Control group ($p < 0.001$), and showed a concentration-dependent trend. Based on the data obtained, the calculated result of the $IC_{50}$ of the $H_2O_2$ solution concentration was 0.375 mmol/L, which was used for the establishment of the HepG2 cellular oxidative stress model.

#### 3.6.3. Effects of SFAJ and IFAJ on the Cell Viability of $H_2O_2$-Induced HepG2 Cells

The protective effect quantified by the cell viability of SFAJ and IFAJ on $H_2O_2$-induced HepG2 cells against oxidative stress is presented in Figure 9a. The cell viability dropped to $50.23 \pm 3.77\%$ in the Model group ($p < 0.001$) compared with $100.00 \pm 0.69\%$ in the Control

group, indicating that $H_2O_2$ successfully induced oxidative stress damage. Compared with the Model group, the cell viability in the SFAJ group increased to $54.53 \pm 0.85\%$ but showed no significant difference ($p > 0.05$), while the cell viability in the IFAJ group increased to $69.89 \pm 0.43\%$ ($p < 0.01$). To a certain extent, SFAJ and IFAJ could alleviate the oxidative stress injury of HepG2 cells caused by $H_2O_2$, and the antioxidant activity of IFAJ was significantly better than that of SFAJ, which might be related to the antioxidant compounds (phenols, flavonoids, and terpenoids); in particular, the inoculation of *Plantilactobacillus plantarum* probably had a promoting effect by elevating the abundance of these bioactive components. The results further indicate the advantages of inoculated fermentation in improving antioxidant activity, and provide theoretical foundations for the preparation of EPJ by inoculated fermentation.

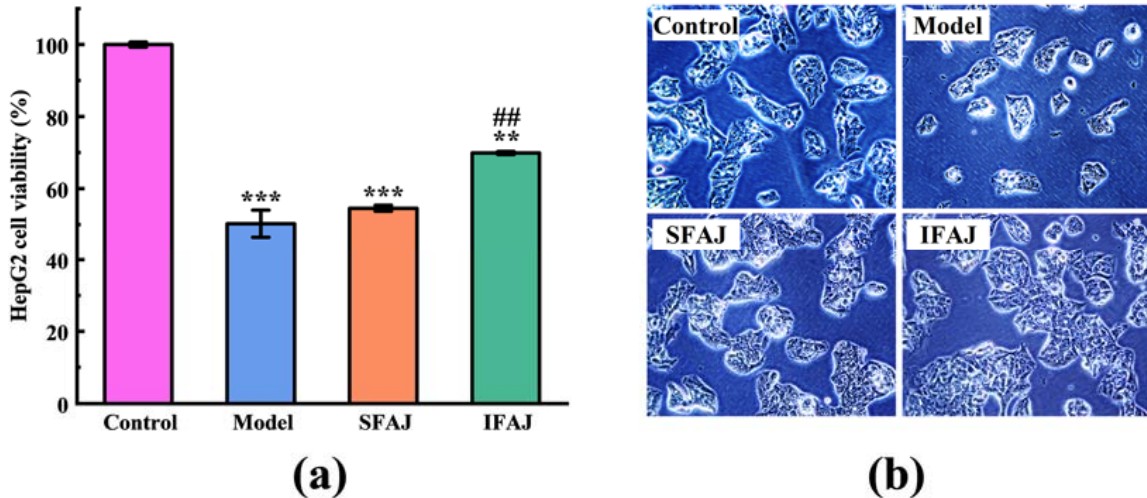

**Figure 9.** The protective effects of SFAJ and IFAJ on the viability (**a**) and morphology (**b**) of $H_2O_2$-induced HepG2 cells. Compared with the Control group, the symbols ** and *** indicate $p < 0.01$ and $p < 0.001$, respectively; compared with the Model group, symbol ## indicates $p < 0.01$.

The morphology of HepG2 cells undergoing different treatments is presented in Figure 9b. A macroscopic decrease in the number, apparent shrinkage, and volume reduction of cells was seen in the Model group compared with that in the Control group, which indicated apoptosis and the morphological changes of HepG2 cells after treatment with $H_2O_2$. In contrast, the SFAJ group and IFAJ group exhibited more normal and morphologically intact cells. Notably, the cells in the SFAJ and IFAJ groups appeared to be more numerous, morphologically plumper, and growing more vigorously compared with the cells in the Control group, suggesting that SFAJ and IFAJ might facilitate the growth and proliferation of HepG2 cells.

3.6.4. Effects of SFAJ and IFAJ on Intracellular ROS Level and the Activities of Antioxidant Enzymes in $H_2O_2$-Induced HepG2 Cells

Excessive ROS, as represented by both free radical and non-free radical oxygenated molecules such as hydrogen peroxide, superoxide, singlet oxygen, and hydroxyl radical, will trigger oxidative stress in the body, leading to chronic and degenerative diseases [51]. The effects of SFAJ and IFAJ on the intracellular ROS level and the activities of antioxidant enzymes in $H_2O_2$-induced HepG2 cells are shown in Figure 10a. Compared with the Control group, the ROS level in the Model group rose to $136.78 \pm 2.82$ U/mg prot ($p < 0.001$), while the ROS level in the SFAJ and IFAJ groups remained at $101.93 \pm 3.25$ U/mg prot and $99.96 \pm 4.07$ U/mg prot ($p > 0.05$), respectively. Correspondingly, the ROS levels in the SFAJ and IFAJ groups decreased by 25.48% and 26.92%, respectively, when compared to those in the Model group ($p < 0.001$). However, the ROS levels in the SFAJ and IFAJ groups were not significantly different from those in the Control group ($p > 0.05$). The results above indicate that SFAJ and IFAJ effectively removed the generated ROS and prevented

its excessive accumulation in cells, thereby keeping the intracellular ROS at a normal level and protecting the cells from oxidative stress damage.

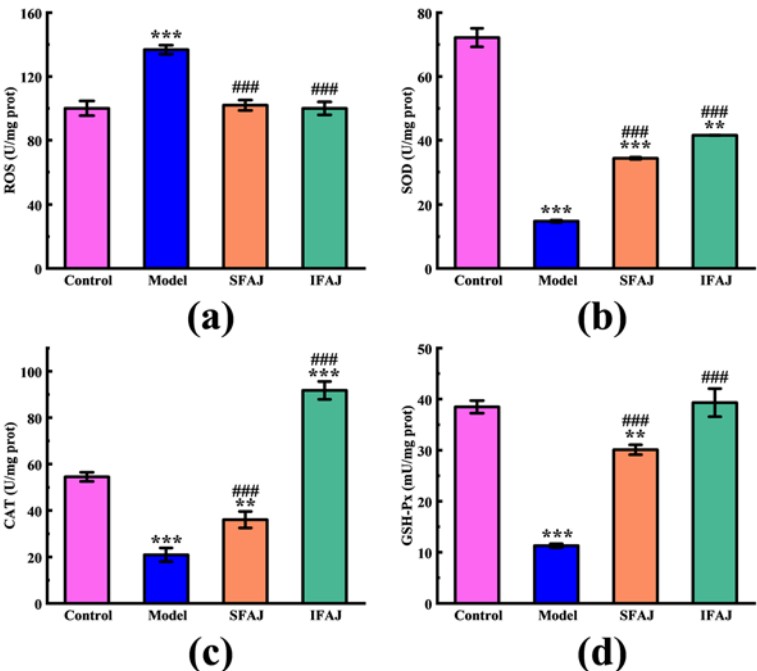

**Figure 10.** The effects of SFAJ and IFAJ on the intracellular ROS level (**a**), the activities of SOD (**b**), CAT (**c**), and GSH-Px (**d**) in $H_2O_2$-induced HepG2 cells. Compared with the Control group, symbols ** and *** indicate $p < 0.01$ and $p < 0.001$, respectively; compared with the Model group, symbol ### indicates $p < 0.001$.

Endogenous antioxidant enzymes are composed of SOD, CAT, GSH-Px, etc., which cooperate to resist oxidative attacks and thus play a crucial role in maintaining the human body in a healthy and stable state [52]. As shown in Figure 10b, the SOD activity in the Model group dropped to $14.80 \pm 0.36$ U/mg prot ($p < 0.001$) compared to $72.18 \pm 2.91$ U/mg prot in the Control group. Compared with the Model group, the SOD activities recovered to $34.31 \pm 0.34$ U/mg prot ($p < 0.001$) and $41.56 \pm 0.06$ U/mg prot ($p < 0.001$) in the SFAJ and IFAJ groups, increasing by 131.82% and 180.81%, respectively. As seen in Figure 10c, in terms of CAT activity, this decreased to $20.91 \pm 2.98$ U/mg prot in the Model group compared with $54.49 \pm 1.96$ U/mg prot in the Control group ($p < 0.001$). Compared with the Model group, CAT activities in the SFAJ and IFAJ groups were $36.06 \pm 3.56$ U/mg prot ($p < 0.001$) and $91.78 \pm 3.85$ U/mg prot ($p < 0.001$), increasing by 72.45% and 338.93%, respectively. As shown in Figure 10d, GSH-Px activity decreased in the Model group to $11.30 \pm 0.38$ mU/mg prot ($p < 0.001$), compared with $38.48 \pm 1.24$ mU/mg prot in the Control group. Compared with the Model group, GSH-Px activities in the SFAJ and IFAJ groups recovered to $30.08 \pm 0.97$ mU/mg prot ($p < 0.001$) and $39.32 \pm 2.75$ mU/mg prot ($p < 0.001$), with increases of 166.19% and 247.96%, respectively. There was no doubt that both SFAJ and IFAJ showed excellent in vitro antioxidant activities. As previous studies have reported, the decline in intracellular ROS levels and the upregulated expression of antioxidant enzymes could be attributed to multiple factors, such as the activation of the TGF-β/Smad signaling pathway [53], positive regulation of the expression of the genes and proteins associated with oxidative stress and autophagy [54], etc. Furthermore, the findings based on the antioxidant enzyme activities showed that IFAJ possessed a better capacity for protecting HepG2 cells from oxidative damage, while the activities of CAT and GSH-Px in the IFAJ group recovered to the level of the Control group, which suggested the better physiological activity of inoculated fermentation by *Plantilactobacillus plantarum*.

## 4. Conclusions

In this study, two types of *Akebia trifoliata* fruit Jiaosu (SFAJ and IFAJ) were fermented and their key metabolites and antioxidant activities were tracked, tested, and compared during the fermentation process. The results of correlation analysis and PCA showed that there were significant positive correlations between phenols, flavonoids, and terpenoids in SFAJ and IFAJ and their antioxidant activities. Network pharmacology was used to elaborate upon the potential anti-oxidative stress mechanism of fermented *Akebia trifoliata* fruit in the treatment of oxidative stress and intuitively verified its effectiveness through HepG2 cells. The results showed that β-sitosterol was probably the key compound in the pharmacological action of *Akebia trifoliata* fruit, while PTGS2, CASP3, CASP8, and CASP9 might be the key proteins in the pharmacological action of fermented *Akebia trifoliata* fruit for SFAJ and IFAJ. Besides, both SFAJ and IFAJ were effective in alleviating the oxidative stress seen in the HepG2 cells. In particular, IFAJ performed better than SFAJ in terms of protecting cells with an intracellular ROS level of 99.96 ± 4.07 U/mg prot, SOD activity of 41.56 ± 0.06 U/mg prot, CAT activity of 91.78 ± 3.85 U/mg prot, and GSH-Px activity of 39.32 ± 2.75 mU/mg prot in the IFAJ group.

In the present study, for the first time, the effect of the improved antioxidant capacity of *Akebia trifoliata* fruit via inoculated fermentation by *Plantilactobacillus plantarum* was confirmed. In addition, a further antioxidant bioactivity study is recommended for the development of novel antioxidative products, which might also provide further information regarding the processing of other fresh fruits or vegetables.

**Supplementary Materials:** The following supporting information can be downloaded at: https://www.mdpi.com/article/10.3390/fermentation9050432/s1, Figure S1: Changes in ethanol content of SFAJ and IFAJ during fermentation.

**Author Contributions:** Conceptualization, R.S. and Z.W.; data curation, Y.S.; formal analysis, Y.S.; funding acquisition, J.M.; investigation, Y.S. and Z.W.; methodology, R.S. and Z.W.; project administration, R.S. and J.M.; resources, Y.C.; software, J.D. and Y.M.; supervision, Z.W.; validation, J.D.; writing—original draft, Y.S.; writing—review and editing, R.S. and Z.W. All authors have read and agreed to the published version of the manuscript.

**Funding:** This study was financially supported by the Fumin Qiangxian Projects of Jiangsu Province (XZ-SZ202027).

**Institutional Review Board Statement:** Not applicable.

**Informed Consent Statement:** Not applicable.

**Data Availability Statement:** Not applicable.

**Acknowledgments:** All the authors are grateful for Huike Li, whose company generously provided researching materials and platform to support our study.

**Conflicts of Interest:** The authors declare that there are no conflict of financial interests or personal relationships in connection with the report submitted.

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
