# Peer review of "Improved Antioxidant Capacity of Akebia trifoliata Fruit Inoculated Fermentation by Plantilactobacillus plantarum, Mechanism of Anti-Oxidative Stress through Network Pharmacology, Molecular Docking and Experiment Validation by HepG2 Cells"

_fermentation, doi:10.3390/fermentation9050432_

Round 1

Reviewer 1 Report

The Manuscript describes new and relevant information about antioxidant capacity of Akebia trifoliata fruit, submitted to spontaneous and inoculated fermentation with Lactobacillus plantarum.

The experimental assays have been well designed, performed, described and interpreted. The multivariate statistical analysis has been well applied and reported in Figures. The conclusions are supported by the experimental and simulation results.

The Manuscript is acceptable for publication in its present version.

Best regards.

Reviewer 2 Report

The manuscript describes the effects of fermentation on the functional quality, especially the free polyhenol and terpenoid content of Akebia trifoliate fruit. both spontaneous fermentation and fermentation induced by Lactiplantibacillus plantarum (former Lactobacillus plantarum) were applied. Both fermentations followed the same overall pattern, but the L. plantarum strain caused a more stable and predictable outcome. Fermentation released phenols and terpenoids, increased the antioxidative activity of the fruit, and protected the cultured caco2 cells from the lethal oxidative damage caused by H2O2. The effects of the bioactive compounds on the gene expression and metabolicroutes were modelled.

The experimental approach is sound, and the results appear reliable. However, the authors should address the following points:

1) Bacterial taxonomy. Lactobacillus plantarum is now Plantilactobacillus palntarum

2) What was the L. plantarum inoculum (as CFU per /ml) during the inoculation, and how did the L. plantarum cell count develop during the fermentation?

3) Was the divelopment of the bacterial, yeast and mold counts in the spontaneously fermented product not monitored?

4) In Figure 5 the legend should be improved. The legend should be self-explanatory, and therefore the meanings of "Akerbia trifoliata fruit", "Oxidative stress" and their overlap in the Venn diagram shouldbe briefly described in the legend.

Although the English of the manuscript is generally understandable, alinguistic revsion, preferably by a native speaker, is needed

Reviewer 3 Report

In my opinion, the manuscript entitled Improved Antioxidant Capacity of Akebia Trifoliata Fruit Inoculated Fermentation by Lactobacillus Plantarum, Mechanism of Anti-Oxidative Stress through Network Pharmacology, Molecular Docking and Experiment Validation by HepG2 Cells, by Sun et al., is a quite interesting and well documented work. The results are discussed and compared with the current state of the art, the materials and method are presented (but need to be better explained), and the conclusion is relevant. 

I have the following comments and suggestions:

1. point 3.3. Why did authors fermented at 25 ± 5 °C, considering that Lactobacillus plantarum strain is fermented at 37 °C, according to a large body of literature such as (Paucean et al. 2013)(Teleky, Martău, and Vodnar 2020)(ChiÅŸ et al. 2020)?

2. How or why authors decided to add pectinase, magnesium sulfate heptahydrate, sodium dihydrogen phosphate? The matrix substrate of Akebia Trifoliate is not able to support a fermentation process?

3. Line 523: which was the initial quantity of Lactobacillus plantarum strain? Why authors decide to add 3.75 g of Lactobacillus plantarum strain? 

4. How Lactobacillus plantarum strain was prepared before the inoculation? Dis authors use MRS medium?

5. The fermentation for 72 days is quite long and difficult to apply in the industrial technology. Generally, in the literature the authors used a maximum days of 3 or 5 for a goos fermentation. Why 72 days?

5. Did authors measured the total acid content or TTA (total titrable acidity)? TTA is usually used to measure the increase of total titrable acidity in fermented samples.

6. which was the formula used for the total acid content conversion? Lines 535-537

7. line 692. Please use improper pronoun – instead of our work please use: in the present study, or this work ….

8. please mentioned in the abstract the abbreviation of ROS, SOD and CAT (lines 22-23).
